# Gynecological Diagnosis and Treatment of Ectopic Ureter Insertion into Vagina: Analysis of Five Cases and a Literature Review

**DOI:** 10.3390/jcm11216267

**Published:** 2022-10-24

**Authors:** Qijing Wang, Zaigui Wu, Fengbin Zhang, Rubab Akbar, Yiyun Lou, Jianhong Zhou, Fei Ruan

**Affiliations:** 1Department of Gynecology, Women’s Hospital, School of Medicine, Zhejiang University, Hangzhou 310006, China; 2Department of Reproductive Endocrinology, Women’s Hospital, School of Medicine, Zhejiang University, Hangzhou 310006, China; 3Department of Pharmacology and Systems Physiology, College of Medicine, University of Cincinnati, Cincinnati, OH 45267, USA; 4Department of Gynecology, Hangzhou Hospital of Traditional Chinese Medicine, Hangzhou 310007, China

**Keywords:** female urology, ectopic ureter, genital tract malformation, urinary tract malformation, urinary incontinence

## Abstract

An ectopic ureter is a ureter that does not correctly connect to the trigone of the bladder and drains outside of the bladder. Here, we presented five cases of ectopic ureter opening into the vagina, whose clinical symptoms and malformations were rarely described in previous case reports. All five patients were hospitalized with complaints of gynecologic disease. Three of the five cases did not present the typical symptoms of urinary incontinence. Three of these cases showed congenital malformations of the female genital tract. Four cases were diagnosed in adulthood. All patients were analyzed using various imaging examinations. This study suggests that the ectopic ureter should be considered in the differential diagnosis of a pelvic mass in a patient with urinary and reproductive system abnormalities. It is essential to comprehensively evaluate complex malformations of the genitourinary system with multiple imaging tests.

## 1. Introduction

An ectopic ureter is a ureter that terminates outside the trigone of the bladder [1]. The overall incidence of ectopic ureter in the population is 0.05–0.025%, with a female-to-male ratio of 2–6:1. Most cases are diagnosed in childhood and rarely in adulthood [1,2]. A previous study over 7 years reported that patients diagnosed with an ectopic ureter ranged in age from 1 to 13 years, with a mean age of 4.5 years [3]. Notably, the uncommon cases diagnosed in adults suggest that the prevalence of ectopic ureter may be higher [4]. Patients with ectopic ureter generally present with other malformations of the urinary system, such as a duplex or single collecting system, even with female genital tract anomalies [5]. The diagnosis of rare malformations of ectopic ureter is mainly conducted based on relevant case reports [2]. Therefore, more information about cases of ectopic ureter needs to be described and discussed. The most common sites for ureter insertion in females are the bladder neck and upper urethra (33%), the vaginal vestibule (33%), the vagina (25%), the uterus (less than 5%), and the cervix (less than 5%) [1]. When the ureter drains into the vagina, incontinence is the main symptom for girls with an ectopic ureter. In addition, the ectopic ureter is also associated with an increased risk of urinary tract infections (UTIs) and hydroureteronephrosis [1]. However, some cases are easy to miss and misdiagnose at the initial visit due to the insidious onset of symptoms of ectopic ureter [6].

In the case of an ectopic ureter combined with an atrophic kidney, it is challenging to diagnose depending on a single imaging examination due to the small-sized and dysfunctional kidney. Various imaging tests, including magnetic resonance imaging (MRI), computed tomography (CT), and urography, are applied to demonstrate the malformations. However, the detection rate and diagnostic value of each imaging test are still uncertain [2,5,7]. Geographical differences exist in the prevalence of ectopic ureters in a single collecting system [3,8], and the cellular and molecular mechanisms generating ureter abnormalities are not fully understood [9]. Before surgical therapy, urogenital tract abnormalities manifest in various ways, making identification and evaluation difficult. Thus, we summarize a series of rare clinical cases at the university teaching hospital that initially presented with gynecological complaints before being identified as an ectopic ureter with vaginal insertion. The diagnosis, imaging characteristics, and clinical management of these cases in the gynecological department are all presented in this study.

## 2. Case Reports

### 2.1. Case 1

A 46-year-old woman was admitted to the department of gynecology, reporting dysmenorrhea for 33 years and heavy menstrual bleeding with dizziness for 2 years. She attained menarche at the age of 13 and had severe dysmenorrhea, which impacted her quality of life. The menstrual cycle was 27–33 days, with menses lasting 12 days. The menstrual volume in the last two years doubled. Since then, she also experienced dizziness and fatigue during her periods and was diagnosed with anemia. Between the ages of 15 and 22, she had persistent spotting or moisture within her underwear, but did not require menstrual pads.

Her medical history revealed that in 1999 and 2002, she delivered two children through cesarean section and then underwent tubal ligation. She had a hysteroscopy and laparoscopy in 2005 to treat a uterine malformation and an ovarian endometrioma, respectively. She consistently took antihypertensive medications and had a 10-year history of hypertension. A gynecological examination showed that the uterus was slightly enlarged, and a small hole, approximately 0.5 cm in diameter, was seen at the nine o’clock direction of the cervix.

A three-dimensional transvaginal ultrasound revealed adenomyosis and a septate uterus (Figure 1). Two cystic masses with thick internal fluid measuring 3.7 × 4.2 × 2.7 cm (Figure 1A) and 7.5 × 3.1 × 2.1 cm (Figure 1B) were seen on the right side of the uterus, respectively. These cystic masses were considered to be a hydroureter with vaginal fistula. A urinary tract ultrasound revealed the absence of a right kidney, as well as a dilated right ureter measuring 11.1 × 1.8 cm with an ectopic opening into the right vagina (Figure 2A). A computed tomography urography (CTU) revealed the absence of the right kidney (Figure 2B). A pelvic MRI showed a cystic mass considered to be a mesonephric cyst on the right side of the pelvis, bladder, and urethra (Figure 3A). A complete septate uterus with multiple fibroids was also found (Figure 3B). The creatinine concentration of the fluid inside the pelvic cyst was measured to be 157 nmol/L via transvaginal aspiration (Figure 4).

The patient had severe dysmenorrhea due to adenomyosis and heavy menstrual bleeding, which led to anemia and negatively impacted her quality of life. Therefore, the patient asked for a hysterectomy. Following a multidisciplinary team conference, it was decided to perform an abdominal hysterectomy on this patient. During the hysterectomy, a 3 cm long oblique vaginal septum was found between the right vaginal fornix and the right vaginal wall and removed. Consequently, the 0.5 cm opening next to the cervix found during the gynecological examination was proved to be a hole in this oblique vaginal septum. After this septum was removed, a 1 cm in diameter ectopic ureteral opening was found in the right vaginal wall adjacent to the right vaginal fornix. The yellow fluid was observed inside the ectopic ureter with the same characteristics as shown in Figure 4. We removed the ectopic ureter to prevent fluid leakage, infection, or malignant transformation. Based on the postoperative pathological results of the specimen (Figure 5), the patient was, finally, diagnosed with adenomyosis, uterine fibroids, abnormal uterine bleeding, uterine septum, oblique vaginal septum, the absence of the right kidney, right ureteral dysplasia, an ipsilateral ectopic ureter opening into the vagina, and a female mesonephric cyst.

### 2.2. Case 2

A 14-year-old girl was admitted to the department of gynecology due to monthly cyclic pelvic pain. A pelvic examination showed an imperforate hymen. A rectoabdominal examination showed a cyst 10 cm in diameter palpated in the pelvis. Additionally, the bottom edge of the cyst was located 1 cm away from the vaginal opening. The needle aspiration fluid through the hymen revealed values of 5213.2 μmol/L for creatinine and 39.63 mmol/L for urea. An abdominal MRI revealed a left pelvic cystic mass, left hematosalpinx, and left kidney atrophy (Figure 6). The pelvic MRI demonstrated that the fluid-filled vagina was highly dilated, and the left dilated ureter drained into the vagina (Figure 7). An ascending urography also revealed a left ectopic ureter. After the incision of the imperforate hymen, 500 mL of pale-yellow purulent pus drained from the vagina. After an antibiotic treatment, a laparoscopy and hysteroscopy were performed. The surgical procedures revealed a double primordial uterus measuring 2.5 × 1.9 cm on the right and 3.0 × 1.5 cm on the left, but no cervix. Bilateral ovaries and fallopian tubes were normal. The patient’s vagina was unobstructed. An opening orifice of the left ureter, with a diameter of 0.2 cm, was seen in the vagina. Subsequently, the left atrophic kidney and the irregularly enlarged left ureter were removed to prevent future leakage and infection. In summary, the left ectopic ureter opened into the vagina, causing fluid leakage into the vagina. Due to the imperforate hymen, urine accumulated inside the vagina, which led to vaginal dilatation, infection, and pelvic pain.

### 2.3. Case 3

A 37-year-old woman was referred to our hospital for persistent vaginal discharge. After she underwent vaginal cyst excision 11 years ago, she began to experience persistent vaginal discharge, requiring two sanitary pads daily. A pelvic examination revealed a pinpoint-like hole in the right anterior wall of the vaginal fornix. It was observed that clear fluid was leaking from the hole, and a 3-centimetre in diameter cystic mass was palpated nearby. An MRU demonstrated that the duplicated renal pelvis was located at the upper pole of the right kidney. The left kidney and ureter were normal (Figure 8A). An MRU also revealed the dilated upper segment, thinned middle segment, and slightly expanded lower segment of the right duplicated ureter. Additionally, the distal segment of the duplicated ureter was enlarged to 1.7 × 1.3 cm, terminating at the right wall of the vagina. (Figure 8B). Consequently, the right ectopic ureter was separated at its base and reimplanted into the bladder.

### 2.4. Case 4

A 29-year-old woman was admitted to the hospital with complaints of urine incontinence since childhood and an ovarian cyst for three years. A urinary tract ultrasound demonstrated a right duplex kidney with an ectopic ureter and left kidney agenesis (Figure 9A). A CTU demonstrated a left ectopic ureter (Figure 9B). Because the patient had previously undergone a laparoscopic ovarian cystectomy for endometriosis and caesarean delivery, peritoneal adhesions potentially caused the laparoscopy to be challenging. Finally, the ovarian endometriosis cyst was removed, and the left ectopic ureter was reimplanted into the bladder via laparotomy.

### 2.5. Case 5

A woman of 33 years old was advised to undergo a hysteroscopy before receiving assisted reproductive technologies. She received a hysteroscopic uterine septum resection at age 27. Her right fallopian tube was removed due to two ectopic pregnancies. A pelvic examination showed an oblique septum extending 3 cm from the left fornix of the vagina to the hymen border. The pelvic ultrasound demonstrated an ectopic ureter or ureterovaginal fistula (Figure 10). An MRI and CTU detected left renal agenesis and a left dilated ureter with an ectopic opening. Following vaginal septum excision, a double cervix with a dysplastic left cervix was found. Given that the left ectopic ureter was asymptomatic, this ectopic ureter was not surgically removed.

### 2.6. Clinical Characteristics of the Five Cases

A summary of the clinical characteristics of each case is shown in Table 1.

## 3. Discussion

In this study, all five cases were admitted to hospital for gynecological-related complaints. The diagnosis of the ectopic ureter was easily missed due to its rare incidence. Urinary incontinence, manifested as persistent wetness of the underwear or urine leakage, was a common symptom of an ectopic ureter with vaginal insertion in women [7]. Case 3 and Case 4 presented urinary incontinence requiring a sanitary pad. Case 1 had continuous moisture within her underwear between the ages of 15 and 22. Case 2 did not present urine leakage due to her imperforate hymen, finally resulting in a vaginal abscess. Therefore, malformations of the urinary system ought to be suspected once we recognize such symptoms.

The diagnostic tests for ectopic ureter were diverse, including ultrasound, CT, MRI, and urography. The diagnostic sensitivity and specificity of each imaging test for ectopic ureter are currently uncertain. In previous case reports, ectopic ureter was diagnosed through an MRU [10], CTU [2,11], or IVP (intravenous pyelogram) [12]. Case 1 was diagnosed with an ectopic ureter using transvaginal and urethral ultrasonography due to a cystic mass at the ectopic vaginal opening of the ureter. In Case 2, the ectopic ureter was diagnosed with both a pelvic MRI and ascending urography. It has been reported that an MRU shows high accuracy in assessing ectopic ureters [13]. In Case 2, the pelvic MRI was confirmed to be the correct diagnosis. However, the abdominal MRI mistakenly interpreted the left dilated ectopic ureter as a left hematosalpinx. An MRU was also applied in Case 3 and correctly diagnosed in Case 3. Excluding the ectopic orifice, abnormalities of the kidney and ureter in all cases were diagnosed using ultrasound.

In the case of a highly suspected ectopic ureter, it was helpful that transvaginal aspiration could be applied to measure the creatinine value of the fluid inside the pelvic cyst near the vaginal wall. In both Cases 1 and 2, the elevated creatinine value of the fluid within the cystic mass indicated the presence of urine within the cyst.

Previous studies reported varying geographical rates of ectopic ureters in dual [8] and single [3] collecting systems. The incidence of ectopic ureter with a vaginal opening in females remains unclear. In this study, four cases associated with a single system were identified, two of which were associated with ipsilateral kidney agenesis.

The surgical treatment strategy for ectopic ureter is based on the urinary system’s function [1,14]. In Cases 1 and 2, a nephrectomy was performed due to the absence of a kidney or an atrophic kidney. Ureteral reimplantation was applied in Case 3 and Case 4, as the kidney of the ectopic side was functional. Case 5 did not undergo surgery because the atrophic kidney was asymptomatic.

The developmental disorders of the urinary system throughout embryogenesis contribute to the ectopic ureter mechanisms [7]. Although the exact mechanisms of ectopic ureter remain unclear, the complex process depends on precise cellular signaling pathways and the integration of diverse progenitor cell types. Kidney development is initiated by the outgrowth of the epithelial ureteric bud from the mesonephric or Wolffian duct in response to signals from the metanephric mesenchyme. This signaling pathway involves several members of the transforming growth factor family (TGF family), including SMAD family member 4 (SMAD4) and bone morphogenetic protein 4 (BMP4). SMAD4 and BMP4 also contribute to the development of the female reproductive system [9,15,16]. The development of the urinary and reproductive systems is closely associated with initial periods of embryogenesis [17,18]. The ureteral bud sprouts from the Wolffian duct and grows cranially towards the metanephric blastema. The connection between these structures activates the signaling pathways that stimulate the creation of renal tissue. As a result, ureteral bud ectopia is often linked to the abnormal development of nephrogenic blastema, which causes patients with single-system ectopic ureters to have a dysfunctional or dysplastic ipsilateral kidney and other genitourinary malformations [15,18]. In the cases of female reproductive system anomalies, such as Cases 1, 2, and 5, the screening test for urinary system abnormalities was recommended. None of the patients in this study showed urine incontinence or persistent vaginal discharge after treatment during follow-up.

## 4. Conclusions

This study reported rare cases diagnosed with ectopic ureter insertion in the vagina. The differential diagnosis of ectopic ureters ought to be considered when a pelvic mass of unknown origin is detected in patients with malformations of the urinary and reproductive systems. This study suggested that a combination of multiple imaging tests ought to be applied to diagnose ectopic ureter. A multidisciplinary team consisting of urologists, obstetricians, and radiologists is necessary to diagnose patients with complex malformations and symptoms. Clinical symptoms, including urinary incontinence, urinary tract infections, and ipsilateral renal function, play an essential role in the surgical strategy for treating ectopic ureter.

## Figures and Tables

**Figure 1 jcm-11-06267-f001:**
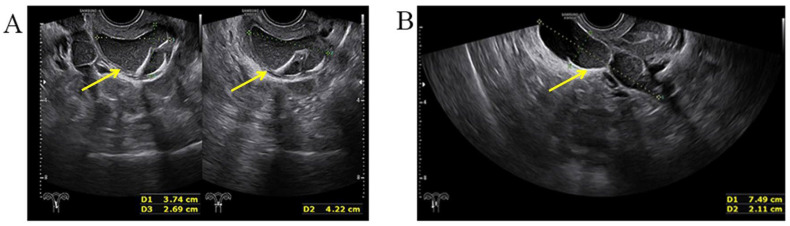
Three−dimensional transvaginal ultrasound. (**A**) Transvaginal ultrasound demonstrated a cystic mass with thick internal fluid measuring 3.7 × 4.2 × 2.7 cm on the right side of the cervix. (**B**) Transvaginal ultrasound demonstrating a cystic mass measuring 7.5 × 3.1× 2.1 cm was seen next to the right side of uterus and its opening into right vaginal wall.

**Figure 2 jcm-11-06267-f002:**
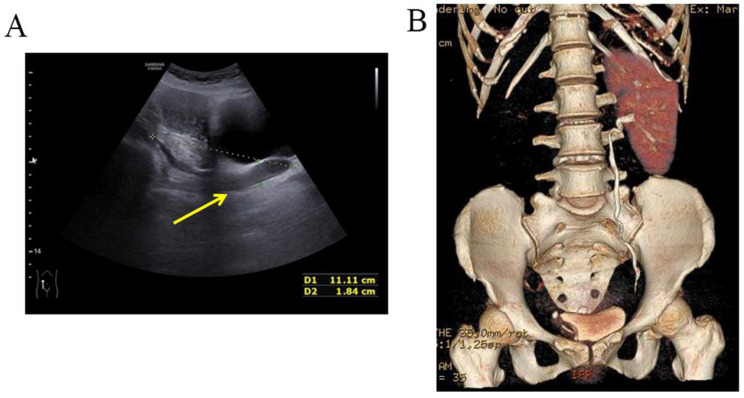
Malformations of the urinary tract. (**A**) Urinary tract ultrasound demonstrated the absence of right kidney, right ureteral dilatation, and right ectopic ureter. (**B**) CTU demonstrated the absence of right kidney.

**Figure 3 jcm-11-06267-f003:**
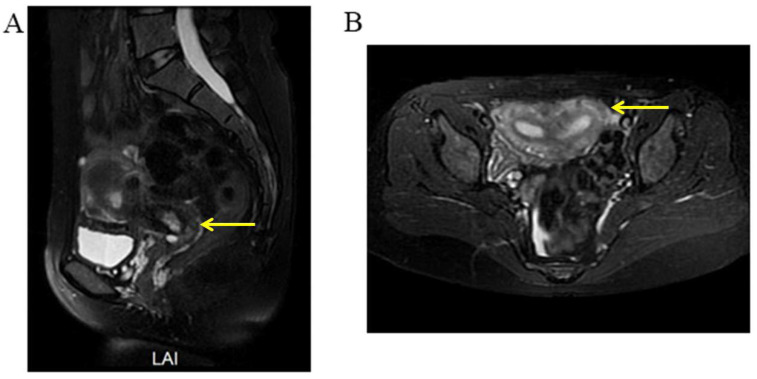
Selected MRI images of the pelvis. (**A**) Pelvic MRI showed a fluid-filled cyst (yellow arrow) along the right side of the pelvis, bladder, and urethra (mesonephric remnants). (**B**) Pelvic MRI showed a complete septate uterus with multiple fibroids (yellow arrow).

**Figure 4 jcm-11-06267-f004:**
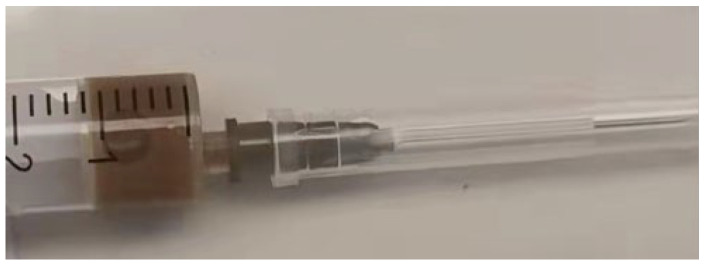
Detection of the creatinine of the fluid inside the pelvic cyst through transvaginal aspiration.

**Figure 5 jcm-11-06267-f005:**
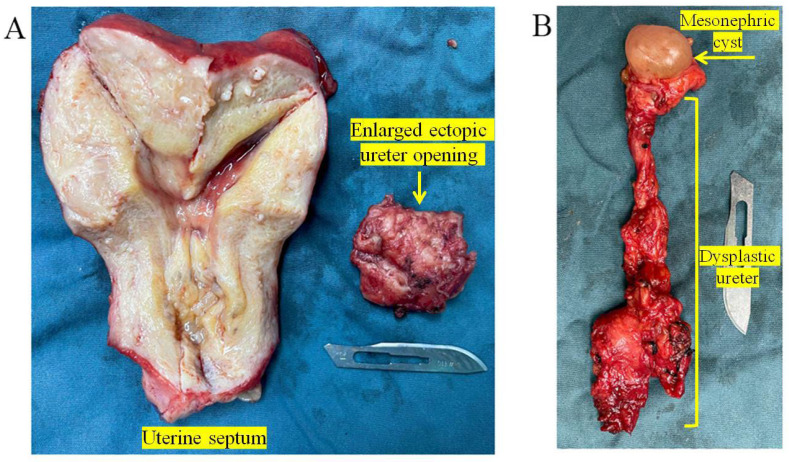
Postoperative specimen. (**A**) Postoperative specimen and pathology showed complete septate uterus with adenomyosis and fibroids. A 4.0 × 4.0 × 3.0 cm ipsilateral enlarged ectopic ureter opening into the vagina. (**B**) A 1.5 × 1.0 × 1.0 cm female mesonephric cyst. A 7.0 × 2.0 × 1.0 cm right dysplastic ureter.

**Figure 6 jcm-11-06267-f006:**
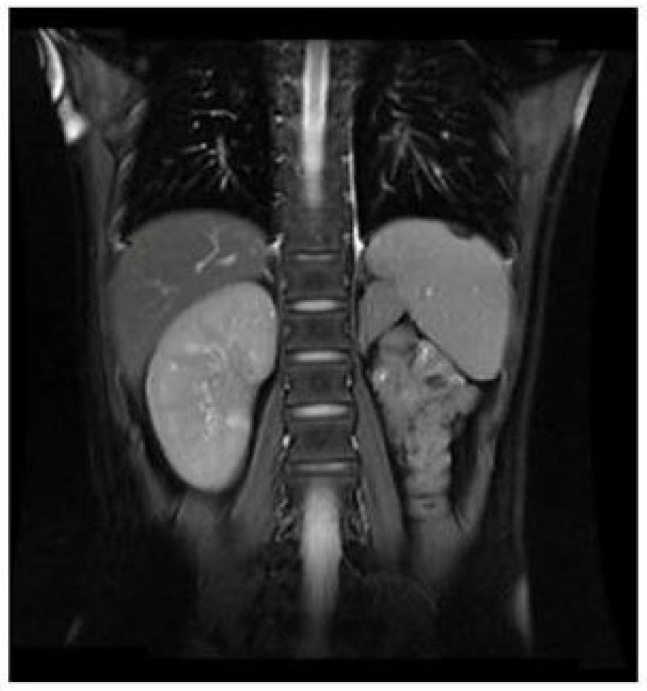
MRI demonstrated the atrophy of left kidney.

**Figure 7 jcm-11-06267-f007:**
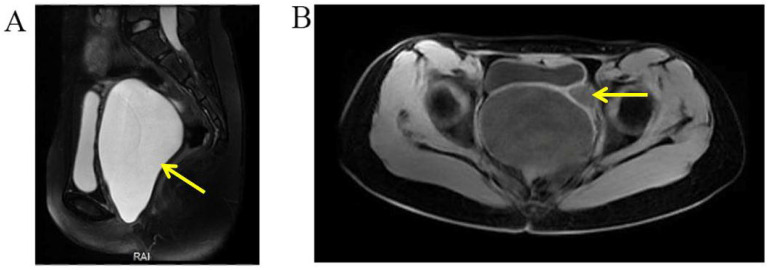
Selected MRI images of the pelvis. (**A**) Pelvic MRI demonstrated a highly dilated vagina with internal fluid measuring 10.2 × 7.3 × 13.2 cm (yellow arrow). (**B**) Left ureter was dilated (yellow arrow), and the ectopic orifice of the left ureter was located at the vagina.

**Figure 8 jcm-11-06267-f008:**
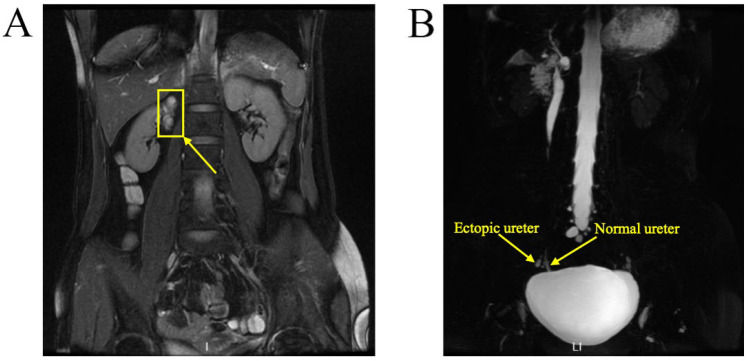
MRU demonstrated a duplex collecting system. (**A**) The duplicated renal pelvis (yellow arrow) was located at the upper pole of the right kidney. (**B**) The duplicated ureter (yellow arrow) drained into the right wall of the vagina.

**Figure 9 jcm-11-06267-f009:**
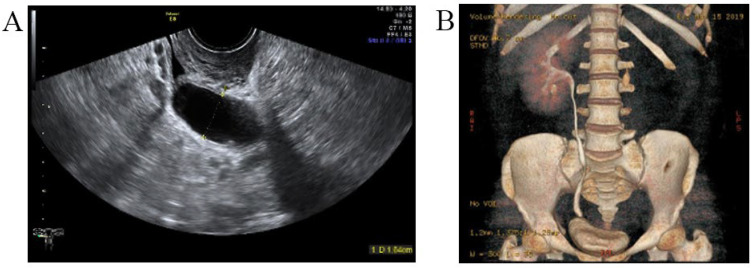
The left ectopic ureter. (**A**) Urinary tract ultrasound demonstrated that the ectopic ureter extended to the middle and upper parts of the vagina, and the widest diameter of this ureter (yellow arrow) was 1.6 cm. (**B**) The CTU demonstrated a left ectopic ureter.

**Figure 10 jcm-11-06267-f010:**
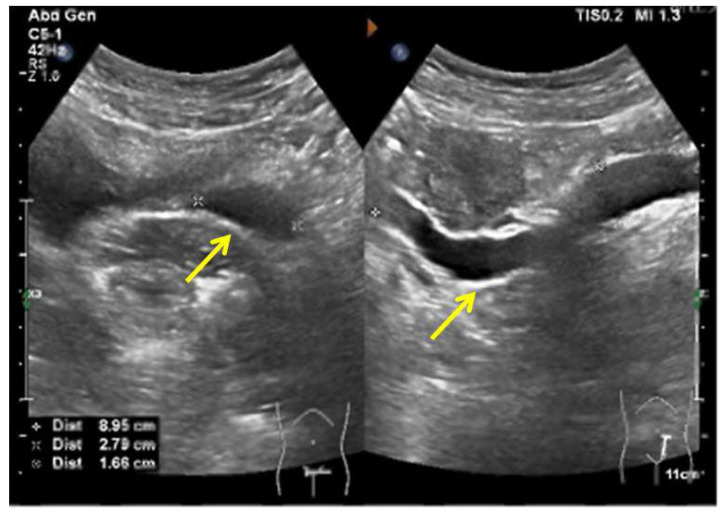
The pelvic ultrasound demonstrated that a sausage-shaped ectopic ureter (yellow arrow) was found behind the uterus and bladder.

**Table 1 jcm-11-06267-t001:** The summary of clinical characteristics of each case.

	Case 1	Case 2	Case 3	Case 4	Case 5
Admission date	2022.03	2018.06	2015.07	2019.12	2020.11
Age (years)	46	14	37	29	33
Gender	female	female	female	female	female
Chief complaint	dysmenorrhea and heavy menstrual bleeding	cyclic pelvic pain	vaginal discharge	urinary incontinence and ovarian cyst	endometrial polyps
Gynecological- related conditions	adenomyosis and menorrhagia	imperforate hymen with vaginal abscess	persistent vaginal discharge	ovarian endometrioma, right hydrosalpinx	recurrent pregnancy loss, endometrial polyps, reproductive duct anomalies
Imaging test	pelvic ultrasound, urinary ultrasound, pelvic MRI, CTU	pelvic ultrasound, urinary ultrasound, pelvic MRI, CTU, ascending urography	pelvic ultrasound, MRU, CTU, pelvic CT, abdominal CT	pelvic ultrasound, urinary ultrasound, pelvic MRI, CTU	pelvic ultrasound, pelvic MRI, CTU
Imaging test demonstrating ectopic ureter with vaginal insertion	pelvic ultrasound, urinary ultrasound	pelvic MRI, ascending urography	MRU	pelvic MRI, CTU	pelvic ultrasound, pelvic MRI, CTU
Congenital malformations of the female genital tract	uterine septum, oblique vaginal septum	double primordial uterus, imperforate hymen	none	none	oblique vaginal septum, double cervix, complete septate uterus
Urinary tract malformation	absence of right kidney, right ureteral dysplasia, ipsilateral ectopic ureter opening into the vagina	left kidney atrophy, left ureteral dilatation, left ureteral ectopic opening into the vagina	right duplex kidney, right ureteral ectopic opening into the vagina	left ureteral ectopic opening into the vagina	left renal agenesis and left dilated ureter with ectopic opening
The orifice site of ectopic ureter	vaginal wall	inside a sac of the vaginal wall	the right anterior wall of vaginal fornix	upper region of vagina	vagina
Surgical treatment of ectopic ureter	ureterectomy	ureterectomy	ureteral reimplantation	ureteral reimplantation	conservative observation
Karyotype	/	46, XX	/	/	46, XX

## Data Availability

Not applicable.

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
