# Peer review of "Gynecological Diagnosis and Treatment of Ectopic Ureter Insertion into Vagina: Analysis of Five Cases and a Literature Review"

_jcm, 2022, doi:10.3390/jcm11216267_

Round 1
Reviewer 1 Report
The current study presents a rare uro-gynecological abnormality, that has a polymorphous clinical presentation and challenging diagnostic process.
Overall, the paper is well put together, with a thorough research protocol.
The introduction describes the incidence of the malformation, underlying its rarity and the difficulty of the evaluation workflow. For this section I would recommend the following data to be added:
- The sex-ratio of encountered ectopic ureter cases.
- Mean age at diagnosis.
- Percentage of patients left undiagnosed.
- Long-term complication rates.
The Materials and methods section elaborates each of the 5 documented cases of ureteric opening malformation. Regarding these cases, I would recommend the following:
Case 1: Although the vaginal opening of the ureter was made clear through multiple investigations, the gynecological exam showed a 5 mm opening on the left side of the cervix. Please clarify this finding.
Case 2: Multiple findings are present in this case’s description, that correlate poorly with one another. It is not clear if the vulvar abscess is the same 10 cm mass palpated at 1 cm away from the vaginal opening, objectified through an MRI evaluation. Please clarify if the vulvar abscess and the imperforated and dilated vagina are two separate structures. Additionally, the aetiology of the left hematosalpinx is not explained, please detail this issue.
Case 3: Please provide MRI images of the right duplex kidney as well as intraoperative pictures of the reimplantation procedure.
Case 4: Please provide intraoperative pictures, if available.
Finally, the Discussions section elaborates the clinical characteristics of each case in Table 1. However, this would be more suitable for the Results. Please consider relocating the table. When debating the sensitivity of the imagistic diagnostic procedures, in the second case, the authors mention that the first diagnostic method was ascending urography, which was not mentioned in the initial case presentation. Please revise this issue. Another minor issue would be the use of the term ‘nephrectomy’ in a patient with agenesis of the renal unit. In that case, I suggest the use of ureterectomy rather than nephrectomy. Finally, the last paragraph links the ectopic ureteral opening to multiple birth defects of the urogenital system. However, the authors do not describe a certain pathological pathway or how the uretero-vaginal fistula may be connected with the agenesia of aplasia of the ipsilateral kidney or to gynecological malformations.
Author Response
Dear Reviewers,
We appreciate the reviewers for your precious time reviewing our paper and providing valuable comments. Your helpful and insightful suggestions led to possible improvements in the current version. We have provided point-by-point responses below and tried our best to address every one of them. We hope the manuscript, after careful revisions, can meet your high standards. The authors welcome further constructive comments, if any. You can track all modifications in the revised manuscript in the attachment.
Response to Reviewer 1 Comments
Point 1: The introduction describes the incidence of the malformation, underlying its rarity and the difficulty of the evaluation workflow. For this section I would recommend the following data to be added:- The sex-ratio of encountered ectopic ureter cases. - Mean age at diagnosis. - Percentage of patients left undiagnosed. - Long-term complication rates.
Response 1: Thank you! We found your comments extremely helpful and have revised them accordingly. Please see lines 32-36 in the manuscript. The literature on long-term complication rates is limited, and the exact data remains unclear. Thank you for pointing out an important research direction we will explore in future studies if possible.
Point 2: Case 1: Although the vaginal opening of the ureter was made clear through multiple investigations, the gynecological exam showed a 5 mm opening on the left side of the cervix. Please clarify this finding.
Response 2: Thank you for this suggestion! We agree that this 5 mm opening should be more accurately described. Please see lines 113-120 in the manuscript. We revised this as follows: The 0.5 cm opening next to the cervix found on the gynecological examination was proved to be a hole in this oblique vaginal septum. After this septum was removed, a 1 cm in diameter ectopic ureteral opening was found in the right vaginal wall adjacent to the right vaginal fornix. We have stated more details to make it more clear.
Point 3: Case 2: Multiple findings are present in this case’s description, that correlate poorly with one another. It is not clear if the vulvar abscess is the same 10 cm mass palpated at 1 cm away from the vaginal opening, objectified through an MRI evaluation. Please clarify if the vulvar abscess and the imperforated and dilated vagina are two separate structures. Additionally, the aetiology of the left hematosalpinx is not explained, please detail this issue.
Response 3: We appreciate your constructive comment. The vulvar abscess palpated on the physical examination was proved to be the fluid-filled dilated vagina. So the vulvar abscess and imperforated and dilated vagina are the same structure. We modified this description in case 2 to make it more exact. Please see lines 132-134, 138-139 and 161-164 in the manuscript.
For the aetiology of the left hematosalpinx, we explained this imaging result in lines 18-20 of the discussion. The abdominal MRI mistakenly interpreted the left dilated ectopic ureter as left hematosalpinx. It indicates that the imaging of malformation is challenging for radiologists.
Point 4: Case 3: Please provide MRI images of the right duplex kidney as well as intraoperative pictures of the reimplantation procedure.
Response 4: Thank you! We have added the MRI images of the right duplex kidney to Case 3 at your suggestion. Please see Figure 8.
We understand that intraoperative pictures can clarify the case's description, but we did not photograph during the operation seven years ago. We are very grateful for your great suggestion. As this is a retrospective study, we are sorry that we cannot provide intraoperative images. With the application of laparoscopy and da Vinci technology, we will preserve intraoperative pictures for sharing and communication in future surgeries.
Point 5: Case 4: Please provide intraoperative pictures, if available.
Response 5: Thank you for your suggestion! We will accept your advice and preserve intraoperative pictures in future surgeries.
Point 6: Finally, the Discussions section elaborates the clinical characteristics of each case in Table 1. However, this would be more suitable for the Results. Please consider relocating the table.
Response 6: Thank you! We have relocated Table 1 to the Result section. Please see page 8.
Point 7: When debating the sensitivity of the imagistic diagnostic procedures, in the second case, the authors mention that the first diagnostic method was ascending urography, which was not mentioned in the initial case presentation. Please revise this issue.
Response 7: Thank you for this excellent observation. We have revised our statement about the results in line 138 of case 2 and line 16 of the discussion.
Point 8: Another minor issue would be the use of the term ‘nephrectomy’ in a patient with agenesis of the renal unit. In that case, I suggest the use of ureterectomy rather than nephrectomy.
Response 8: Thank you so much for catching this error. We have modified the word accordingly. Please find this in Table 1.
Point 9: Finally, the last paragraph links the ectopic ureteral opening to multiple birth defects of the urogenital system. However, the authors do not describe a certain pathological pathway or how the uretero-vaginal fistula may be connected with the agenesia of aplasia of the ipsilateral kidney or to gynecological malformations.
Response 9: Thanks for your suggestion to make the statement on mechanisms of the ectopic ureter more detailed. We have carefully revised this part according to your insightful comments. Please find it in the last paragraph in the Discussion section.

Reviewer 2 Report
jcm-1956230
The authors present 5 female patients who were found with an ectopic ureteral orifice that either caused or not symptoms.
The structure is correct but some questions were raised while studying it.
Introduction:
32. “An ectopic ureter is a ureter that terminates outside the trigone of the bladder [1]. It is reported that the incidence rate of the ectopic ureter in autopsies is 1900:1”. Probably, the authors wanted to write 1:1900. Please correct it.
Case 1:
103-104 The authors tell us that they did the surgery in order to prevent urine leakage. Since there is no kidney on the right, where would the leakage come from?
Finally, it is a well prepared study. It is true that the ectopic ureteral orifice is not something usual, but the diagnosis and therapy are standard. So, this study didn’t offer something new.
Author Response
Dear Reviewers,
We appreciate the reviewers and editors for your precious time reviewing our paper and providing valuable comments. Your helpful and insightful suggestions led to possible improvements in the current version. We have provided point-by-point responses and tried our best to address every one of them. We hope the manuscript, after careful revisions, can meet your high standards. The authors welcome further constructive comments, if any. You can track all modifications in the revised manuscript.
Response to Reviewer 2 Comments
Point 1: 32. “An ectopic ureter is a ureter that terminates outside the trigone of the bladder [1]. It is reported that the incidence rate of the ectopic ureter in autopsies is 1900:1”. Probably, the authors wanted to write 1:1900. Please correct it.
Response 1: Thank you so much for catching this error. We have modified the word accordingly. Please find it in line 32.
Point 2: Case 1:103-104 The authors tell us that they did the surgery in order to prevent urine leakage. Since there is no kidney on the right, where would the leakage come from?
Response 2: Thank you so much! During the surgery, we found a one cm-in-diameter ectopic ureteral opening in the right vaginal wall adjacent to the right vaginal fornix. The yellow fluid was observed inside the ectopic ureter with the same characteristics as shown in Figure 4. Although the kidney was absent, there was fluid inside this enlarged ureter. Thus, we removed the ectopic ureter to prevent fluid leakage, infection, or malignant transformation. Please find this in lines 113-120.
Point 3: Finally, it is a well prepared study. It is true that the ectopic ureteral orifice is not something usual, but the diagnosis and therapy are standard. So, this study didn’t offer something new.
Response 3: Thank you! Although the diagnosis and therapy are standard, these cases were hospitalized for gynecological-related complaints, which have not been seen in previous studies. The study of these cases can encourage more gynecologists to become aware of the ectopic ureter as a differential diagnosis for urinary incontinence or pelvic infection. At the same time, four cases diagnosed in adulthood were rarely reported. Finally, we hope more people will see our study to help patients with ectopic ureters get timely diagnosis and treatment.
